# Maillard Reaction Induced Changes in Allergenicity of Food

**DOI:** 10.3390/foods11040530

**Published:** 2022-02-12

**Authors:** Jingkun Gou, Rui Liang, Houjin Huang, Xiaojuan Ma

**Affiliations:** School of Public Health, Zunyi Medical University, Zunyi 563000, China; gjk20220123666@163.com (J.G.); liangrui0316@163.com (R.L.); houjinhuang123@163.com (H.H.)

**Keywords:** allergen, food allergy, maillard reaction

## Abstract

Food allergy is increasing in prevalence, posing aheavier social and financial burden. At present, there is still no widely accepted treatment for it. Methods to reduce or eliminate the allergenicity of trigger foods are urgently needed. Technological processing contributes to producing some hypoallergenic foods. Among the processing methods, the Maillard reaction (MR) is popular because neither special chemical materials nor sophisticated equipment is needed. MR may affect the allergenicity of proteins by disrupting the conformational epitope, disclosing the hidden epitope, masking the linear epitope, and/or forming a new epitope. Changes in the allergenicity of foods after processing are affected by various factors, such as the characteristics of the allergen, the processing parameters, and the processing matrix, and they are therefore variable and difficult to predict. This paper reviews the effects of MR on the allergenicity of each allergen group from common allergenic foods.

## 1. Introduction

Food allergy (FA) is an adverse immunological reaction to dietary foods, involving dermal, respiratory, and gastrointestinal discomforts. For severe cases, anaphylaxis and even death may occur [1]. The prevalence of FA and common allergen types vary from country to country, as they are affected by many factors, such as age, genetics, ethnicity, and dietary habits. FA occurrence rates in the general population in the US, Canada, and Australia are 6.7%, 6.1%, and 11%, respectively [2,3]. The prevalence of FA among children in some developing countries is also high, at 4.3% in Turkey (for preschool children at a mean age of 28 ± 6 months) and 7.7% in China (for children 0–24 months old) [4,5]. To make it worse, the incidence of FA has been rising over the past few years [6,7]. FA not only significantly impacts patients’ lives and psychology but also imposes a heavy financial burden on families and society. FA has become a serious global public health concern.

Food processing might work as an effective way to change the fate of an allergen in triggering symptoms. Among all the processing methods, heating is the most commonly used, as foods are usually consumed after cooking. When natural or added sugars are involved in a heating process, the Maillard reaction (MR) may occur. It is a non-enzymatic browning reaction between the amino acid residue of protein and reducing sugar, also known as glycation [8]. Sites for MR are free amino groups on protein (lysine residues, arginine residue, and amino acid on the N-terminal in protein). MR is a complicated process that can be roughly divided into three stages: an early stage, an advanced stage, and a final stage. The three stages are interrelated and can occur simultaneously depending on reaction conditions [9]. The MR may destroy, mask, or modify sequential epitopes of an allergen by introducing sugar chains, and conformational epitopes may also change, thus affecting the allergenicity of food (Figure 1).

This paper reviews the effects of the MR on the allergenicity of common allergenic foods from a food science point of view. There have been several reviews concerning the MR and allergenicity of foods; however, they either include various processings, and MR is just a minor part containing limited information [10,11], or they are written from a more clinical perspective [8,12,13]. Moreover, because the MR occurs during heating, this review not only describes MR-induced changes in allergenicity but also differentiates it from heating-induced changes.

## 2. Definition of Indicators Usedto Determine Food Allergenicity

In FA diagnosis and allergenicity-related food processing research, the oral food challenge (OFC) should be the gold standard. However, in food science and food chemistry, performing the OFC on patients is not that applicable. In practice, given that the most common type of FA is IgE-mediated, which involves the mechanism of IgE-FcεRI-mast cell/basophil axis, the IgE binding ability test is usually introduced [14]. Under most circumstances, the IgE reactivity test in food research uses in vitro assays such as ELISA and Western blot [10]. Nevertheless, allergens after ingestion would confront a series of events such as degradation and digestion, so finally it is not the original allergens ingested that crosslink with IgE. Thus, in vitro IgE reactivity of an allergen, widely used in food research, is more appropriately a reference that indicates potential allergenicity. To mimic the whole ingestion process, including the contribution of the gastrointestinal mucosa system, FA animal models can be used, where serum-specific IgE level is commonly used as an indicator for allergenicity in vivo. However, a higher serum-specific IgE level is not always consistent with mast cell/basophil degranulation and the onset of symptoms, as it may not yet arrive at the threshold [15]. Therefore, evaluation of allergenicity in vivo is usually accompanied by allergy symptoms, histamine level, mast-cell protease-1 (MCP-1) level, and a Th2 bias in CD4^+^ T cells [16,17].

To avoid misunderstanding, the indicators used in this review are defined following the work of Costa et al. [11]. IgE binding capacity (or IgE reactivity) refers to the ability of an allergen to bind to IgE in the sera of FA patients. Allergenicity in vivo is confined to the ability to elicit allergy in animal models. In vitro gastrointestinal digestion is another important component to describe allergenicity change in food science. A digested allergen after the MR usually shows increased IgE binding capacity because the glycation sites (lysine and arginine residue) are also the cleavage sites of the digestive enzyme. However, the IgE binding ability of the digested allergen is not equal to allergenicity because the peptides would then confront antigen processing. We did not address more work on this due to the amount of information included in IgE binding capacity and allergenicity in vivo, and it has been reviewed in other works [13,18]. 

## 3. Maillard Reaction and Allergenicity of Foods

It has long been accepted that more than 90% of food allergies are caused by eight food sources: eggs, milk, peanuts, wheat, shellfish, soy, fish, and tree nuts [19]. However, in the summary report of the ad hoc Joint FAO/WHO Expert Consultation on Risk Assessment of Food Allergens issued in May 2021, sesame was included on the allergenic food “big eight” list, substituting soy, although it is still a food source that induces allergy regionally [20]. There are also some other allergenic foods that have a high prevalence in certain parts of the world; for example, buckwheat is in the top eight allergenic foods in Japan, and allergy to fruit is on the list in South Asia [21,22]. In the following, the world-wide, well-recognized allergic foods and MR-induced changes in allergenicity are discussed.

### 3.1. Milk

Cow’s milk allergy (CMA) is the most frequent cause of FA in infants and young children, with a reported prevalence of 1–3% worldwide at pre-school age [23,24,25]. CMA puts a financial burden on families and health services all over the world; moreover, it draws much attention due to the importance of cow’s milk and dairy products in infants’ daily life. Cow’s milk contains a rich variety of proteins, with the major allergens including caseins and β-lactoglobulin and α-lactalbumin in whey protein [26]. 

During milk production, under high temperatures such as in pasteurization, glycation will naturally happen without lactose elimination [27]. In a work mimicking the spray-drying process, β-lactoglobulin was treated with or without lactose. The result showed that at an air inlet temperature of 120 °C and air outlet temperature of 60 °C, the IgE binding capacity of β-lactoglobulin was only reduced when treated with lactose; at an air inlet temperature of 180 °C and air outlet temperature of 90 °C, its IgE binding capacity was reduced in the absence of sugar due to aggregation, and glycation led to a further decrease [28]. However, our literature review showed that the possibility of glycation reducing the IgE binding capacity of β-lactoglobulin compared with the heating process seems to be related to the sugar type, reaction condition, and detection sera used. For example, when ribose, fructose, maltose, or arabinose was present, glycation further reduced the IgE binding capacity of β-lactoglobulin compared to heating, but reaction with lactose did not cause much difference because lactose induced a much lower glycation degree, and those glycated sites were not on main epitopes [29,30]. Under some dry and wet heating conditions, individual difference in the sera used even determines the increase or decrease in the IgE binding capacity [31]; see Table 1 for more detailed reaction conditions. Reaction with polysaccharides reduced the IgE binding capacity of β-lactoglobulin. Wu et al. [32] found that instead of inducing apparent conformational changes, the reduction in β-lactoglobulin IgE binding capacity after glycation was mainly caused by the blocking of epitopes, which are on the surface of the allergen molecule. Besides IgE reactivity reduction, glycation reduced the allergenicity of β-lactoglobulin in vivo by reducing its intestinal epithelial transfer, increasing its uptake by bone-marrow-derived dendritic cells and speeding up its degradation there [33]. All these changes significantly alter β-lactoglobulin’s fate as an allergen.

In consideration that β-lactoglobulin has both linear and conformational epitopes, there are efforts to pretreat the allergen before glycation for the purpose of promoting allergenicity reduction. In fact, ultrasound and sonication pretreatment did show a synergic effect with glycation to reduce the IgE binding capacity of β-lactoglobulin because the pretreatment not only helps to change the conformation of the allergen (especially in increasing α-helix) but also increases the number of glycation sites [34,35,36]. The IC50 of its IgE binding capacity combining ultrasound pretreatment and glycation could be reduced to 20.07 μg/mL, compared with 2.67 μg/mL of the native protein, 2.6 μg/mL of the heated control, and 7.89 μg/mL of the glycated sample without pretreatment at 400 W.

For the other reported allergen in whey protein, α-lactalbumin, glycation was also proved to reduce its IgE binding capacity, from an IC50 of 5.73 μg/mL to 10.79 μg/mL (a reduction more than heating control), due to both conformational changes and a glycation-introduced sugar chain that was located on epitopes [37]. There are also some reports that combined pretreatment with glycation in the control of α-lactalbumin allergenicity. The results showed that both pretreatment with ultrasound and high-pressure microfluidization further reduced the IgE binding capacity of the allergen compared with samples glycated without pretreatment [38,39]. 

For whey protein isolate, glycation with dextran was reported to reduce its IgE-binding capacity. However, the reduction rate of IgE binding reactivity did not correlate with the protein surface coverage, and it is likely to be caused by the greater surface steric hindrance of the reacted molecule [40]. The isolate was also predigested before glycation. The whole process led to a 99% reduction in its IgE binding capacity [41]. Interestingly, the final sample (whey protein isolate hydrolyzed by immobilized trypsin and chymotrypsin and then glycated with dextran) also greatly decreased bitterness compared with the digested protein. The above evidence shows that glycation under proper conditions reduced the IgE binding capacity of single whey protein and whey protein isolate more than sole heating. Thus, adding reducing sugar in whey protein production when a heating period is involved is probably useful in terms of allergenicity control.

Glycation was reported to change the aggregation state and conformation of casein, exerting different effects compared with heating. However, their corresponding allergenicity changes were seldom investigated, and the only relative research we found provided data that its IgE binding capacity was preserved [42,43].

### 3.2. Egg

Eggs are one of the most common allergic foods in infants and young children, and the most important type is hen’s egg (HE) allergy, with a prevalence estimated to be between 1% and 2% for children aged <5 years, and for infants at age 1 year, it can reach as high as 9.5% [44,45]. Besides daily consumption, egg ingredients are extensively used in the food industry, giving a greater chance of accidental exposure [46]. The main allergens of HE are recognized as ovomucoid, ovalbumin, ovotransferrin, and lysozyme in egg white.

Ovalbumin accounts for 54% of egg white protein, while it is not the immune-dominant allergen and its allergenicity is heat vulnerable [47,48,49]. However, heating in the presence of sugar would not always reduce its IgE binding capacity. Yang et al. [50] reported that when the allergen is incubated in the presence of sugar, its IgE binding capacity is lower than the heated control because glycation induced drastic structural changes to destroy its conformational epitopes, and some linear epitopes were blocked. Ma et al. [51] performed glycation of ovalbumin, and the result indicated that the process increased the IgE binding capacity of the allergen due to the exposure of the hidden epitopes. There were some attempts to reduce the IgE binding capacity of ovalbumin by glycation combined with some pretreatment. It seems that the preheating (60 °C for 1 h, lower preheating temperature) did not show a good effect, while a higher temperature would affect the reconstitution of the allergen, and ultrasound pretreatment (400 W or 600 W for 15 min) helps to further reduce the IgE binding capacity of ovalbumin after glycation, especially when 600 W ultrasound pretreatment was included. The IC50 increased from 10.03 μg/mL of the glycated protein to 28.02 μg/mL of the final sample (ovalbumin glycated with mannose following ultrasound pretreatment at 600 W; IC50 of the native ovalbumin was 3.10 μg/mL) [52,53]. In consideration of its allergenicity change in vivo, glycation with mannose was proven to be a good choice [54]. The glycated ovalbumin significantly prevented high IgE, Th2 cytokines, histamine concentration, and the frequency of allergic signs in the mice model.

Ovomucoid is the immune-dominant allergen in egg white. In the work of Jiménez-Saiz et al. [49], glycation was performed on ovomucoid, producing an ovomucoid–glucose conjugate with an increased IgE binding capacity. No more research was found on glycation and ovomucoid allergenicity. Patients with persistent egg allergy tend to have IgE binding antibodies to ovomucoid linear epitopes, indicating that the allergy to ovomucoid has individual differences. Thus, a more precise conclusion is that glycation increased the IgE binding reactivity of ovomucoid for the sera studied. It was reported that long-term glycation changed both secondary and tertiary structures of lysozyme, and even non-structural aggregates were formed [55]. Glycation was also reported to change the conformation of ovotransferrin [56]. Those changes in structure may influence the allergenicity of a protein. However, we found no research on the effects of glycation on the allergenicity of ovotransferrin and lysozyme. In terms of egg white, glycation reduced its allergenicity in vivo [57]. 

### 3.3. Peanut

Peanut allergy is very common in Western countries, with an estimated prevalence of approximately 0.2–5.2% in the general population [58,59]. It is a serious and sometimes fatal FA type, more severe than other types of FA, with a much lower chance than egg and milk allergy to outgrow. There are up to 13 allergens in peanuts, of which Ara h 1, Ara h 2, Ara h 3, and Ara h 6 are the most important [60,61]. There is already much evidence that heat treatment affects the allergenicity of peanut allergens [62,63]. Because the content of reducing sugar in peanuts is comparatively high, MR may occur during heating.

It is generally believed that roasting increases the allergenicity of peanut proteins, while boiling reduces it, although many researchers attributed the reduction by boiling to the transfer of small allergens to the boiling water [64]. The changes in peanut allergenicity during these processes are relevant to glycation. Maleki et al. [65] performed glycation on Ara h1, Ara h2, and whole peanut extracts (WPE) with various sugar types. Results showed increased IgE binding capacity after glycation. Gruber et al. [66] investigated the glycation of Ara h2 and concluded that its IgE binding capacity increase was due to sugar conjugation onto non-basic amino acids in major epitopes. However, during our literature review, we found that, for individual peanut allergens, glycation did not always increase its IgE reactivity. Peanut allergens Ara h 1 and Ara h 2/6 were respectively roasted with glucose (H + G) or without glucose (H-G). It was found that all the treated samples had decreased IgE-binding ability compared to the corresponding untreated proteins. Moreover, compared to H-G samples, glycation further decreased the IgE-binding activity of Ara h 1 due to the conjugation of the sugar chain and reduced the accessibility rendered by large aggregation [67]. Furthermore, whether glycation induced a reduction in peanut allergen IgE reactivity compared with the boiling control is affected by individual differences of peanut allergy patients [68,69]. When these glycated peanut allergens were used in the mice model, an increased allergenicity in vivo was reported, possibly due to the formation of advanced glycation end products (AGE) [70,71].

Peanut allergen Ara h 1 and Ara h 3 were reported to form AGE during peanut roasting, which is more difficult to digest, facilitating increased allergenicity in vivo. Moreover, AGE can activate the receptor on the antigen presenting cell, which may result in its presentation of antigens to T cells, increasing its sensitizing capacity [12,71]. The AGE of peanut was also found to increase the degranulation capacity of RBL cells and the IgE binding capacity in vitro, indicating that a neoepitope was formed, or the existing epitope had a higher ability to bind with IgE [71,72]. Although a greater focus is on AGE-induced allergenicity changes in peanut allergens, other food allergens such as fish tropomyosin are also reported to form AGE during the glycation process [73,74]. 

Frying was also reported to reduce the IgE binding capacity of peanut allergen Ara h 1 and Ara h 2, although the reduction is not as great as boiling [75,76]. The mice model showed a reduced allergenicity of Ara h 2 in vivo [76]. Conformational changes were detected and used to explain the allergenicity results. However, whether or not glycation contributes to the allergenicity change was not investigated in these works.

### 3.4. Shellfish

Shellfish is a big family and includes more than 50,000 species of crustaceans and more than 100,000 species of mollusks, but the most allergy-causative species are shrimps, crabs, lobsters, clams, oysters, and mussels [77]. Shellfish consumption varies by region, and in some regions, the prevalence of shellfish allergy is high. A cross-sectional study including 3864 school children (11–14 years old) in Kuwait found that 1.3% of children had shellfish allergy, just below egg and fish allergy percentages [78]. In some coastal countries, allergic reaction to shellfish was a predominant FA [79]. The most important shellfish allergens are reported to be tropomyosin, arginine kinase, myosin light chain, sarcoplasmic calcium-binding protein, and troponin C [80].

Glycation-induced changes in tropomyosin allergenicity seem to be source-dependent. Most of the works in this field are targeted at shrimp tropomyosin. According to Zhang et al. [81], the IgE-binding capacity of shrimp tropomyosin was decreased after glycation. The same result was supported by the work of Fu et al. [82], where tropomyosin reacted with either ribose, galacto-oligosaccharide, or chitosan-oligosaccharide. Samples from all the three test systems gave up to a 60% reduction in IgE-binding ability. The group also found that it was changes in the α-helix structure that mediated the allergenicity decreases. After glycating shrimp tropomyosin with glucose, maltose, maltotriose, maltopentaose, and maltoheptaose, all the samples showed significant reductions in IgE reactivity, except for tropomyosin-maltose, despite the fact that maltose could highly glycate tropomyosin, possibly due to the formation of AGEs as neoallergens [83]. Apart from IgE-binding reductions, glycation with tropomyosin was found to reduce allergenicity in vivo by reducing its sensitization [84,85,86]. However, the sugar type for glycation should be chosen to avoid neoallergen generation that offsets allergenicity reduction [73,74]. Nevertheless, because shellfish is a big family, glycation may cause increases in allergenicity in tropomyosin from some species. For example, glycation with glucose, ribose, and maltose was reported to increase the IgE reactivity of tropomyosin from scallops [87]. An interesting phenomenon for tropomyosin is that conformational changes seem to be more important for tropomyosin allergenicity, and sugar conjugation may only affect the allergenicity of the allergen when the glycation site is located in epitopes [82,85,87,88].

There are some other allergens in shellfish where MR resulted in allergenicity changes. Arginine kinase was glycated with ribose, arabinose, galactose, glucose, and maltose, but only arabinose reduced its IgE-binding capacity. The arginine kinase-arabinose MR product was also found to reduce its allergenicity in vivo [85]. Another important allergen, sarcoplasmic-calcium-binding protein, showed a significant reduction in IgE activity after glycation [89]. Glycation of myosin was reported to affect the microstructure and thermal stability of the allergen, but the articles did not involve changes in its allergenicity [90]. 

### 3.5. Fish

The exact prevalence of fish allergy is difficult to investigate. At present, the reported incidence for a general population varies from 0% to 5%, depending on the diet habits, diagnosis criteria, and age of the population [80,91,92]. Apart from oral intake, fish allergy may also result from the respiratory inhalation of fish aeroallergens or through contact with the skin. Dermal fish allergy incidents are higher in fish processing workers, which are reported to be approximately 2–8% with bony fish alone [93]. Fish allergens mainly include parvalbumin, β-enolase, and Aldolase [94].

Parvalbumin has received more attention than other fish allergens. Zhao et al. [95] found that MR with glucose can reduce the IgE reactivity of recombinant silver carp parvalbumin, because glycation sites overlapped with K88, K97, and K108 in epitopes. Zhang et al. [96] glycated Alaska Pollock parvalbumin with glucose, fructose, ribose, lactose, and galactose, while only the reaction with ribose/galactose reduced its IgE binding capacity. Yang et al. [97] processed parvalbumin by boiling, ultrasonic treatment, ultraviolet treatment, pressure treatment, and MR. Results showed that only the MR reduced its IgE reactivity, and more alleviation was achieved when combined with pressure. The combination of the MR and pressure treatment also reduced parvalbumin allergenicity in vivo. However, IgE-binding of codfish parvalbumin was found to increase after glycation due to the formation of multimers [98]. 

β’-component, a major allergen in chum salmon roe, was purified and processed with MR, and with this method, inhibition of human IgE-binding activity was lower than 50% [99]. Glycation was reported to enhance the bioactivity of fish gelatin, but its allergenicity change was not assessed [100]. 

### 3.6. Tree Nuts

“Tree nuts” is a broad term used to classify nuts, including almonds, cashew, hazelnuts, pistachio, walnuts, and some less-popular nut types. Currently, approximately 0.2–3.3% of the world’s general population is allergic to tree nuts [101,102]. Tree nuts are a heterogeneous group, so for a single food, it is much higher; for example, for 164 Omani patients with allergic reactions, 34.1% showed IgE-binding ability to mixed tree nuts [103]. It is worth noting that approximately 90% of tree nut allergies can persist over a patient’s lifetime. Tree nut allergies also increase the risk of fatal anaphylaxis, and a systematic review found that 55% to 87% of deaths caused by FA are related to peanuts and tree nuts [104]. The major tree nut allergens are vicilins (7S globulins), 2S albumins, legumins (11S globulins), profilins, heveins, and lipid transfer proteins [105]. 

A study focused on MR and the allergenicity of Cor a 11, the major allergen in hazelnuts. MR was undertaken with glucose at different parameters. The results showed that although, under certain parameters, a reduction in IgE reactivity was detected, the heating control reduced more [106]. Thus, the MR may not be a good way to reduce the allergenicity of Cor a 11. Hazelnut protein extraction glycated with glucose reduced the intensity of a band in the position of Cor a 1, but the allergenicity results of this protein were not presented [107]. 

One study reported that MR reduced the IgG reactivity of almond 11S legumin [108]. Although the paper did not report its IgE reactivity, the IgG antibody used (4C10) targets a conformational epitope that overlaps with IgE-binding epitopes [109], so it may also affect its IgE binding capacity.

### 3.7. Wheat

For other food types, repeated occurrence of disorders after ingestion may easily be related to an allergic reaction, but this is not true for wheat. Disorders to wheat include wheat allergy, non-celiac gluten/wheat sensitivity, and wheat-dependent exercise-induced anaphylaxis. Thus, the confirmation of wheat allergy needs to be more professional, especially in the case of wheat-dependent exercise-induced anaphylaxis, which may not occur after every wheat intake and needs the clinician’s ability to suspect the disease in diagnosis [110,111]. A systematic review concerning the Asia–Pacific region reported a wheat allergy prevalence of <0.001% to 0.37% in the healthy general population. For patients with anaphylaxis, the rate range is from 4.3% (for the general population) to 26.1% (for children below 14 years), indicating the importance of wheat allergy in anaphylaxis in the Asia–Pacific region [112]. The reported incidence of wheat allergy in Europe was higher, at 3.6% for the overall age group, according to a systematic review [113]. Major wheat allergens include α-Amylase/trypsin inhibitor (Tri a 28 and Tri a 29.01), ω-5-gliadin, α-purothionin (Tri a 37), and nonspecific lipid transfer protein (Tri a 14) [111].

Wheat flour is an important component of many baked goods, and during the baking process, wheat protein may undergo the MR because sugars are usually present. However, reports on the allergenicity change of wheat proteins after glycation are rare. The fact that most allergic reactions caused by wheat occur after the consumption of baked or extensively cooked wheat products may indicate that wheat products still preserve allergenic potential after the MR [114], but more data are needed to support this conclusion.

### 3.8. Sesame

Sesame is widely consumed in western countries in the form of sesame oil, paste, or whole sesame seed. Its reported prevalence ranges from 0.1% to 0.93% in the general population, and in a study involving children (5–14 years old) with IgE-mediated FA, sesame allergy was reported to affect 17% of them [115,116,117,118,119]. Allergy to sesame usually begin before 2 years of age, and only approximately 20% of them can outgrow it [120]. To make it worse, an estimated 23.6% to 37.2% of patients experienced a severe sesame-allergic reaction [115]. Sesi 1 to Sesi 8 are the allergens in sesame identified, and Sesi 4 and Sesi 5 were reported to cause severe reactions [116]. 

Studies on the allergenicity change of sesame allergen after MR were not found. It is now clear that roasting, a key step to improve the flavor and color of sesame seeds, accompanies the occurrence of the MR [121]. Moreover, cross-reactivity was reported to happen between peanut, tree nuts, and sesame seeds due to structural similarities and shared IgE binding regions [122]; thus, glycation, which affects the allergenicity of peanut and tree nut allergens, might also change the allergenicity of sesame allergens. Hopefully, the recognition of sesame’s importance in FA will promote its allergenicity-related research.

### 3.9. Soy

Soy allergy occurs in up to 0.5% of children (13–15 years old), and most of them can outgrow the allergy symptoms [123]. Gly m Bd 60K, Gly m Bd 30k, Gly m Bd 28k, and Gly m 6 are major soy allergens [124,125,126].

Glycation was found to reduce the IgE-binding activity of Gly m 6 compared with both the native form and the heated control due to changes in the allergen structure [127,128]. Babiker et al. [129] conjugated galactomannan with acid-precipitated soy protein. The dot immunoblotting results indicated a significant reduction in IgE-binding ability, especially for the protein of 34 kDa (Gly m Bd 30 K), which was invisible after the sugar conjugation. When chitosan was added for glycation, soy protein showed an even lower IgE-binding capacity compared with the soy protein-galactomannan conjugation, where the IgE-binding activity of Gly m Bd 30K was completely masked [130]. We found no report on soy protein glycation and changes in allergenicity in vivo.

**Table 1 foods-11-00530-t001:** Effect of MR on the allergenicity of food allergens.

Food	Protein	Glycation Condition	Allergenicity Change	Ref
Milk	β-lactoglobulin	Wet or dry heating with lactose	IgE reactivity similar to heating control, depending on sera used	[31]
		Dry heating with fructo-, galacto-, and isomalto-oligosacharides	Reduced IgE reactivity	[32]
		Dry heating with galactose. Sonication pretreatment assisted	Sonication assisted glycation to reduce its IgE reactivity	[34]
		Dry heating with mannose. Ultrasound pretreatment assisted.	Ultrasound pretreatment assisted glycation to reduce its IgE reactivity	[35]
		Dry heating with ribose. Ultrasound pretreatment assisted	Ultrasound assisted glycation to reduce its IgE reactivity	[36]
		Dry heating with fructose	Glycation further reduced its IgE reactivity than heating	[30]
		Dry heating with arabinose	Glycation but not heating reduced IgE reactivity.	[29]
		Spray drying with lactose	Glycation further reduced its IgE reactivity than heating	[28]
	α-Lactalbumin	Dry heating with galactose	Glycation further reduced its IgE reactivity than heating	[37]
		Dry heating with lactose. High-pressure microfluidiser pretreatment assisted	High-pressure microfluidization assisted glycation to reduce its IgE reactivity	[38]
		Dry heating with galactose. Ultrasonic pretreatment assisted.	Ultrasonic pretreatment assisted. glycation to reduce its IgE reactivity	[39]
	Whey protein isolate	Wet heating with dextran	Reduced IgE reactivity	[40]
Egg	Ovalbumin	Dry heating with glucose, mannose, allose, galactose, and idose	Reduced IgE reactivity compared with heating control	[50]
		Wet heating with glucose	Increased IgE reactivity	[51]
		Dry heating with ribose. Preheating: 60 °C for 1 h.	Preheating assisted glycation to reduce IgE reactivity	[52]
		Dry heating with mannose. Pretreating with ultrasound.	Ultrasound pretreatment assisted glycation to reduce IgE reactivity	[53]
		Dry heating with mannose, glucose, fructose, and ribose	Glycation with mannose reduced allergenicity in vivo compared with heated control	[54]
	Ovomucoid	Dry heating with glucose	Reduced IgE reactivity	[49]
	Egg white	Dry heating with mannose	Reduced allergenicity in vivo	[57]
Peanut	Ara h 1, Ara h 2 or whole peanut extract	Wet heating with fructose, glucose, arabinose, mannose, xylose, galactose, or dextrose	IgE reactivity increased	[65]
	rAra h 2	Wet heating with maltose, glucose, fructose, or ribose	IgE reactivity increased	[66]
	Ara h 1; 2S albumins containing both Ara h 2 and Ara h 6 (Ara h 2/6)	Dry heating with glucose	Glycation reduced IgE reactivity of Ara h 1, but not Ara h 2/6	[67]
	r-Ara h 1	Dry heating with glucose	Reduced allergenicity in vivo	[71]
Shellfish	Tropomyosin from *Exopalaemon modestus*	Dry heating with glucose	Reduced IgE reactivity	[81]
	Tropomyosin from *Exopalaemon modestus*	Dry heating with glucose, maltose, maltotriose, maltopentaose, or maltoheptaose	Reductions in IgE reactivity except tropomyosin- maltose	[83]
	rTropomyosin	Dry heating with ribose, galacto-oligosaccharide, or chitosan-oligosaccharide	Reduced IgE binding capacity	[82]
	Tropomyosin from *Scylla paramamosain*	Wet heating with ribose, arabinose, galactose, glucose, or maltose	Reaction with galactose, glucose, and arabinose showed reduced IgE reactivity. Tropomyosin-arabinose showed reduced allergenicity in vivo	[85]
	Tropomyosin from *Penaeus aztecus*	Dry heating with glucose, maltose, maltotriose, maltopentaose, or maltoheptaose	Reduced allergenicity in vivo excepted when glycated with maltose	[84]
	Tropomyosin from *Exopalaemon modestus*	Dry heating with glucose	Reduced allergenicity in vivo	[86]
	Tropomyosin from *Exopalaemon modestus*	Dry heating with Fructo-, galacto-, mannan-oligosaccharides, or Maltopentaose	Reduced allergenicity in vivo except reaction with Fructo-oligosaccharide	[73]
	Tropomyosin from *Exopalaemon modestus*	Dry heating with glucose, maltose, or maltotriose	Reduced allergenicity in vivo except reaction with maltose	[74]
	Tropomyosin from *Patinopecten yessoensis*	Dry heating with ribose, glucose, maltose, or maltotriose	No change or increased IgE reactivity	[87]
	Arginine Kinase from *Scylla paramamosain*	Wet heating with ribose, arabinose, galactose, glucose, and maltose	Only arabinose reduced the IgE reactivity and allergenicity in vivo	[85]
	rSarcoplasmic-calcium-binding protein	Wet heating with xylose	Reduced IgE reactivity	[89]
Fish	Parvalbumin from *Decapterus maruadsi*	Wet heating with glucose in the autoclave sterilizer	Pressure assisted glycation to reduce IgE reactivity and allergenicity in vivo	[97]
	rParvalbumin	Dry heating with glucose	Reduced IgE reactivity	[95]
	Parvalbumin from *Alaska Pollock*	Dry heating with glucose, fructose, ribose, lactose, and galactose	Reduced IgE reactivity when reacted with ribose/galactose	[96]
	Parvalbumin from *Gadus morhua*	Wet heating with glucose	stronger IgE reactivity	[98]
	β’-component from *Pseudosciaena crocea*	Dry heating with glucose	Reduced IgE reactivity	[99]
Tree Nuts	Cor a 11	Dry heating with glucose	IgE reactivity not reduced more than heating	[106]
Soy	Gly m 6	Dry heating with lactose	Reduced IgE reactivity compared with heating	[127]
	Acid-precipitated soy protein	Dry heating with galactomannan	Reduced IgE reactivity, especially for Gly m Bd 30 K	[129]
	Acid-precipitated soy protein	Dry heating with galactomannan or chitosan	Reduced IgE reactivity, especially for Gly m Bd 30 K	[130]

## 4. Conclusions

The MR is a process that occurs naturally and is greatly accelerated by heat. The reaction cannot be separated from heat treatment, which also involves damage of allergen structure. However, the MR sometimes produces conjugation with conformation different from heating, which affects allergenicity. The MR may also block linear epitopes of an allergen, and neoallergens such as AGEs can also be formed during the process. Our review compared the allergenicity of food allergens after the MR with both the native form and heated control, and only those lower than both were treated as a reduction. In this sense, the MR may work as a good way to reduce the allergenicity of milk (parameters controlled) and soy. However, studies found that the MR is not a reliable way to reduce the allergenicity of peanuts. Fish and shellfish are a large family, and the effect of the MR seems to be species-dependent. For egg, tree nuts, and wheat, more data are needed to draw a conclusion.

## Figures and Tables

**Figure 1 foods-11-00530-f001:**
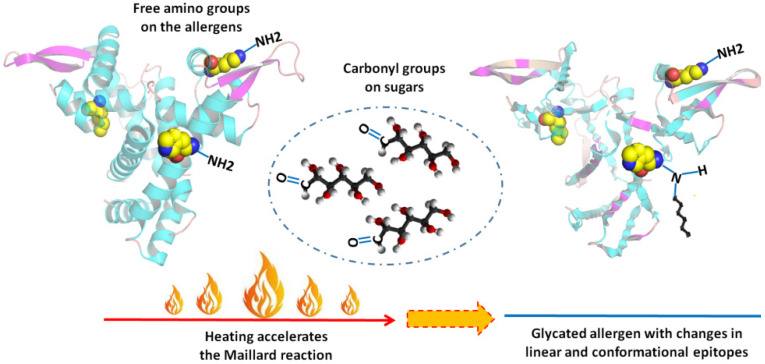
Maillard reaction and epitopes of allergens.

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
