# Peer review of "Maillard Reaction Induced Changes in Allergenicity of Food"

_foods, 2022, doi:10.3390/foods11040530_

Round 1
Reviewer 1 Report
The review is well written and the overview concise and comprehensive.
Major:
* I would just strongly suggest to modify title with deleted "perspective" (there is none in the article) and
* write the abstract in a more neutral way (e.g. effect of MR on allergenicity" instead of positive formulations like "may reduce allergenicity" or "methods to reduce allergenicity" because this is by far not the case for all food allergens.
* AGE needs to be discussed, i.e. separate the stages of MR regarding allergenicity of food products or at least discuss general impact of AGEs in allergic diseases.
Minor:
- add to all prevalence numbers (%) that you cited, which age group of which population is meant (e.g. children <5 years, general population, adult population etc.)
- "MCP-1 level and a Th2-bias: Th2 in what - cells, cytokines, surface markers...?
- paragraph of milk: stress more that pre-digestion (in parallel or additional to high-pressure and ultrasound) has a major impact
- what are "some glycation-inducing agents"?
- paragraph egg: "reduced specific IgE level" is only correct if there have been high IgE-levels before and the ovalbumin was used as a therapeutic treatment, otherwise better use "induced lower IgE" or "prevented high IgE"
- during whole text: "final sample" was often used, please always describe again in detail what that is (e.g. in milk, egg...)
- Ovomucoid: it is mentioned that individual sera might react differently - would that be a fact true for all allergens? then a general statement about reducing or enhancing allergenicity would not be able for any method (e.g. MR)
- paragraph peanut: does MR take place during boiling? when using "heated" in this paragraph, please indicate whether boiling or roasting of peanuts is meant; "AGE selectively interacting with the glycated allergen, thus increasing its sensitization" should be "sensitizing capacity" - and discuss more how that does happen (which receptor/s on which cell/s and what effect/s?)
- paragraph fish: what is bioactivity of fish gelatin?
- paragraph wheat: "confirmation of wheat allergy needs to be more professional" what does that mean? examples?
- paragraph soy: "allergens identified in the blood serum" - is that for sure the allergens or are these specific antibodies to allergens?
Typos: lactaoalbumin, untrasound,
Reviewer 2 Report
The authors present their manuscript reviewing major food allergens and the affect of heat, specifically Maillard reaction.
Section 3, top 8 food allergens is completely based on prevalence of western countries. For example, buckwheat allergy is in the top 8 in Japan. Also, sesame is probably much high in the top eight and probably should not be considered new. Tree Nuts is a heterogenous group (like shellfish), so for a single food it is much higher.
Section 3.1 should start with a run down of major allergens in milk which actual change with the heat and which are more important in clinical allergy prior to the long segments on B-lactoglobin, a-lactalbumin etc.
Section 3.3 should mention the effect of frying.
